# Explaining Metastable Cooperation in Independent Multi-Agent Boltzmann Q-Learning—A Deterministic Approximation

## Abstract

Multi-agent reinforcement learning involves interacting agents whose learning processes are coupled through a shared environment. This work introduces a new discrete-time approximation model for multi-agent Boltzmann Q-learning that accounts for agents' update frequencies. We demonstrate why previous models do not accurately represent the actual stochastic learning dynamics while our model can reproduce several complex emergent dynamic regimes, including transient cooperation and metastable states in social dilemmas like the Prisoner's Dilemma. We show that increasing the discount factor can prevent convergence by inducing oscillations through a supercritical Neimark–Sacker bifurcation, which transforms the unique stable fixed point into a stable limit cycle. This analysis provides a deeper understanding of the complexities of multi-agent learning dynamics and the conditions under which convergence and cooperation may not be achieved.

## 1 Introduction

A typical approach to multi-agent reinforcement learning (MARL) (Albrecht et al., 2024) is to extend single-agent algorithms such as Q-learning to multi-agent settings by treating each agent's learning as an isolated process, called "independent learning" (Tan, 1997) because the processes are only indirectly linked to other agents' learning processes via the shared environment. Although this implies a loss of stationarity on which single-agent convergence guarantees rely (Hernandez-Leal et al., 2017), independent learning is popular for its adaptability and scalability (Matignon et al., 2012) and serves as a competitive baseline in MARL (Papoudakis et al., 2020).

Still, agents' stochastic learning interactions can lead to complex emergent dynamics that can be analysed by dynamical systems theory (Sato et al., 2002; Barfuss & Mann, 2022) if the stochastic algorithms are approximated by deterministic dynamical equations. Often, a continuous-time limit is taken that links the model to evolutionary game theory (Börgers & Sarin, 1997; Sato & Crutchfield, 2003; Galla, 2009). For Q-learning, this was attempted in Tuyls et al. (2003) and later extended to general temporal-difference learning with batches in Barfuss et al. (2019). However, these works do not approximate standard Q-learning but variants with markedly different dynamics. This fact was noted by some (Leslie & Collins, 2005; Kaisers & Tuyls, 2010; Bloembergen et al., 2015; Hernandez-Leal et al., 2017; Barfuss, 2022) but not all works in the field (Kianercy & Galstyan, 2012; Galstyan, 2013; Leonardos & Piliouras, 2022; Mintz & Fu, 2024).

In this work, we clarify the relationships between those approximation models and the actual Q-learning algorithm, explaining the observed discrepancies, and propose a more accurate approximation model. We then use it to explain why (i) rather than converging, independent Q-learning might exhibit stable oscillations due to a moving-target problem, and why (ii) agents often appear to "learn" to spontaneously cooperate over extended periods in social dilemmas where such behaviour is not a Nash equilibrium but merely a metastable phase of the dynamics. We restrict our analysis to a paradigmatic example, the Prisoner's Dilemma, highlighting how even in simple environments, much caution is needed when interpreting learning results.

## 2 BACKGROUND, PROBLEM AND PITFALLS

### 2.1 INDEPENDENT Q-LEARNING IN A SINGLE-STATE ENVIRONMENT

We study a minimal multi-agent system, serving as a paradigmatic example: two agents interacting in a single-state environment, playing the Prisoner's Dilemma, characterised by a single pure Nash equilibrium where both agents defect. In each of finitely many steps, the agents can either choose to cooperate ($C$) or to defect ($D$). The per-step reward tensor is given by

$$\mathbf{R} = \left( \begin{array}{cc} R^1_{CC}, R^2_{CC} & R^1_{CD}, R^2_{CD} \\ R^1_{DC}, R^2_{DC} & R^1_{DD}, R^2_{DD} \end{array} \right) = \left( \begin{array}{cc} 3,3 & 0,5 \\ 5,0 & 1,1 \end{array} \right). \tag{1}$$

At each time step $t$, agent $i$ chooses an action $A^i(t) = a^i \in \mathcal{A}^i$, where $\mathcal{A}^i = \{C, D\}$ and receives a reward $R^i_{\mathbf{A}(t)}$ based on the joint action $\mathbf{A}(t) = (A^i(t), A^{-i}(t))$, where the superscript $-i$ denotes the opponent.

Agents update their Q estimates independently using (Watkins & Dayan (1992), Appendix A)

$$Q^i_{a^i}(t+1) = Q^i_{a^i}(t) + \alpha^i \delta_{A^i(t)a^i} \left( R^i_{\mathbf{A}(t)} + \gamma^i \max_{b^i \in \mathcal{A}^i} Q^i_{b^i}(t) - Q^i_{a^i}(t) \right), \tag{2}$$

where $\alpha^i \in [0, 1)$ is $i$'s *learning rate,* $\gamma^i \in [0, 1)$ the *discount factor,* and $A^i(t) \sim \pi^i(t)$ the random action process governed by a Boltzmann softmax policy $\pi^i_{a^i}(t)$ with *temperature* $T^i > 0$,

$$\pi^i_{a^i}(t) := f(Q^i(t), a^i) = \exp(Q^i_{a^i}(t)/T^i) / \sum_{b^i \in \mathcal{A}^i} \exp(Q^i_{b^i}(t)/T^i). \tag{3}$$

The dynamics of the learning process are fully described by the 4D state vector in Q space, $\mathbf{Q}(t) := (Q^1_C(t), Q^1_D(t), Q^2_C(t), Q^2_D(t))$. The joint policy $\boldsymbol{\pi}(t)$ is a function of $\mathbf{Q}(t)$ that has only two free dimensions due to normalisation, here represented by the cooperation probabilities $\boldsymbol{\pi}_C(t) = (\pi^1_C(t), \pi^2_C(t))$.

We will see that a pitfall for forming an approximate model is that at each time, only the $Q$ values of the actions actually taken are updated, represented by the Kronecker delta $\delta_{A^i(t)a^i}$ in equation 2.

### 2.2 PREVIOUS DETERMINISTIC MODELS

The stochastic nature of MARL makes its dynamics obscure and difficult to interpret (Hernandez-Leal et al., 2017; 2019). The goal of approximation models of MARL processes is to transform them into deterministic dynamical equations that are easier to analyse and reduce the dynamics to its core.

**Frequency-Adjusted Q-learning (FAQL) Model** After Börgers & Sarin (1997) had established a connection between *Cross Learning* (Cross, 1973) and evolutionary game theory, Tuyls et al. (2003)—and similarly Sato & Crutchfield (2003)—attempted to apply their approach to independent Q-learning in single-state environments. The idea is to assume vanishing time steps $\Delta t \to 0$ and a learning rate $\alpha' = \alpha \Delta t$. Tuyls et al. (2003) proposed that the Boltzmann policy in independent Q-learning can thus be approximated in the continuous-time limit by the replicator equation

$$\frac{d}{dt}\pi^i_{a^i}(t) = \alpha^i \pi^i_{a^i}(t) \left( \frac{1}{T^i} \mathbb{E}_{A^{-i}(t)} \left( R^i_{a^i A^{-i}(t)} - \sum_{b^i \in \mathcal{A}^i} R^i_{b^i A^{-i}(t)} \right) + \sum_{b^i \in \mathcal{A}^i} \pi^i_{b^i}(t) \ln \frac{\pi^i_{b^i}(t)}{\pi^i_{a^i}(t)} \right). \tag{4}$$

But in their derivation, they implicitly assumed that *all* Q-values are updated at each time step, effectively treating the update rule equation 2 as if the Kronecker delta $\delta_{A^i(t),a^i}$ was absent. This mistaken assumption has two significant effects: (i) it reduces the dynamics to the two- (instead of four-) dimensional policy space and (ii) makes it independent of the discount factor. They lead to crucial discrepancies between model and actual dynamics (Kaisers & Tuyls, 2010).

Interestingly, their approximation model (which we thus call the FAQL model here) *does* align with a *modified* variant of Q-learning, called *frequency-adjusted* Q-learning (FAQL) (Leslie & Collins, 2005; Kaisers & Tuyls, 2010), which turned out to often be more stable than plain Q-learning. It smooths the learning process by scaling the learning rate $\alpha^i$ with the inverse of the update frequency $\pi^i_{a^i}$, capped at some value $\beta^i$.

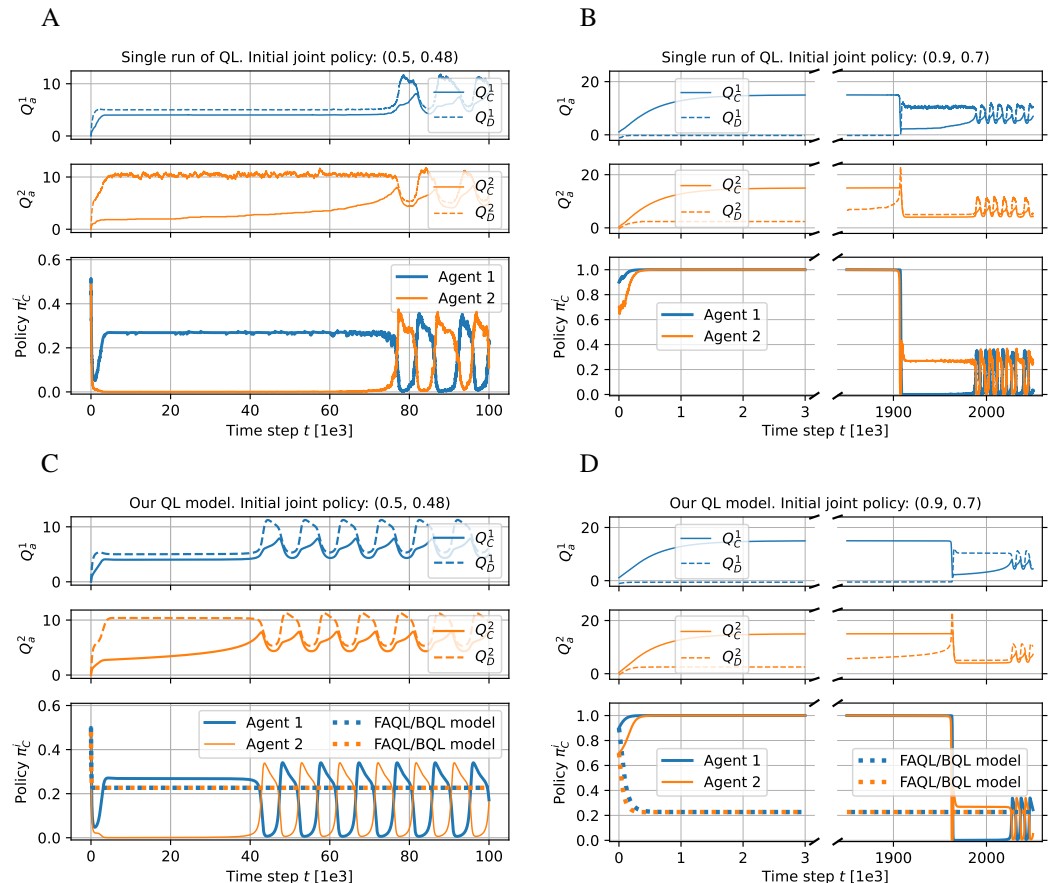

Figure 1: Comparison between a single run of independent Q-learning on the Prisoner's Dilemma (top panels: A, B) and our deterministic approximation model (bottom panels: C, D), defined by equation 6, for $T = 1, \alpha = 0.01, \gamma = 0.8, Q_{base} = 0$. (Timings and trajectories vary across different runs as learning is a stochastic process). We show the evolution of the $Q$-values $(Q_C^1, Q_D^1, Q_C^2, Q_D^2)$ and the resulting probabilities of cooperation $(\pi_C^1, \pi_C^2)$. Dotted policy lines in C and D correspond to previous approximation models FAQL equation 4 and BQL equation 11, obviously fitting the actual dynamics badly. The left panels (A, C) depict an initial joint policy $(\pi_C^1, \pi_C^2) = (0.5, 0.48)$, corresponding to $Q$-values $(0, 0, -0.04, 0.04)$ via equation 5. The right panels (B, C) show an initial joint policy $(\pi_C^1, \pi_C^2) = (0.9, 0.7)$, corresponding to $Q$-values $(1.1, -1.1, 0.4, -0.4)$ via equation 5.

**Batch Q-Learning (BQL) Model**   Barfuss et al. (2019) extended previous deterministic models of MARL to encompass *multi-state* environments with discounting. Their BQL model approximates a *batch* version of temporal difference learning, where the timescales of interaction and adaptation are separated. Appendix B translates the definitions from Barfuss et al. (2019) to our single-state setup, in which the FAQL model corresponds to the continuous-time limit of the discrete-time BQL model; hence, we also collectively refer to them as the 'FAQL/BQL model'.

Both previous approximation models, FAQL and BQL, exhibit the following key characteristics:

1. A fixed point of the dynamics is a boundedly rational strategic equilibrium.

2. They operate within the lower-dim. policy space rather than the higher-dim. Q value space.

3. For single-state environments, they are independent of the discount factor $\gamma^i$.

How well do these models still capture the core dynamics of actual independent Q-learning?

### 2.3 COMPARISON BETWEEN THE FAQL/BQL MODEL AND INDEPENDENT Q-LEARNING

Actual Q-learning occurs in Q space while the FAQL/BQL models operate in policy space, so we need to translate between them. Since $\pi_C^i = 1/(1 + \exp(\Delta Q^i/T^i))$ with $\Delta Q^i := Q_D^i - Q_C^i$, each joint policy corresponds to a 2D affine subspace of Q space. As we want to study the influence of initial conditions in policy space, $\boldsymbol{\pi}(0)$, on the dynamics, we need to translate those to corresponding initial conditions in Q space, $\mathbf{Q}(0)$, which we choose to be

$$Q_C^i(0) := Q_{base}^i - \Delta Q^i(\pi_C^i(0))/2, \quad Q_D^i(0) := Q_{base}^i + \Delta Q^i(\pi_C^i(0))/2, \tag{5}$$

where $Q_{base}^i$ is a parameter that governs the overall initial level of $Q^i$-values. For simplicity, we consider $\alpha^i, \gamma^i, T^i, Q_{base}^i$ to be homogeneous, and omit their indices thereafter.

Figure 1.A and 1.B depict the time evolution of single runs of independent Q-learning for $Q_{base} = 0$, $\gamma = 0.8$ and two different initial joint policies. In both cases, after the first few hundred time steps, the policy trajectories settle into metastable phases where they remain for an extended period. The dynamics then undergo a drastic shift, transitioning into a sustained oscillatory pattern that persists indefinitely. For the initial joint policy $(\pi_C^1(0), \pi_C^2(0)) = (0.5, 0.48)$, this occurs after approximately 70 thousand steps with $\alpha = 0.01$. For $(\pi_C^1(0), \pi_C^2(0)) = (0.9, 0.7)$, the shift is even more pronounced. Initially, the policies seem to converge on mutual cooperation, which appears to contradict individually rational behaviour. However, after about *two million* steps, the trajectories also fall into the indefinite oscillations. In stark contrast, the FAQL/BQL models predict fundamentally simpler behaviour, predicting convergence to a joint policy within just a few hundred steps.

A comparison of the dynamics in policy space highlights the differences. Figure 2.I shows averaged policy trajectories of Q-learning over five runs for two different initialisation approaches and two different values of $\gamma$. Figure 2.II displays the dynamics of the simplified deterministic models. Apparently, the actual trajectories deviate from the model predictions. For $Q_{base} = \min(\mathbf{R})/(1 - \gamma) = 0$, the trajectories follow the edges of the policy space. For $Q_{base} = \max(\mathbf{R})/(1 - \gamma)$, the trajectories initially cluster near the center of the policy space. For $\gamma = 0$, although the trajectories differ from the FAQL/BQL model, they eventually equilibrate around the same fixed point, regardless of initialisation. However, for $\gamma = 0.8$ the trajectories fall into indefinite oscillations, which are not centred around the fixed point. In figure 2.B, some trajectories appear to converge to mutual cooperation in the depicted time span of $1 \times 10^5$ steps. However, as mentioned previously, these states are only metastable. Given sufficient time, the trajectories eventually transition to the oscillatory pattern. Notably, these metastable phases do *not* occur for trajectories initialised at $Q_{base} = 25$.

In summary, the stylised discrepancies are:

1. Whereas the FAQL/BQL model dynamics converge to a single Logit Quantal Response equilibrium in the Prisoner's Dilemma after a couple of hundred steps, actual independent Q-learning does *not* necessarily converge to any strategic equilibrium and may instead settle into oscillations that might emerge only after millions of steps.

2. Whereas the FAQL/BQL model reside in the lower-dim. policy space, actual independent Q-learning dynamics *cannot* be reduced from the higher-dim. Q space: the initialisation ($Q_{base}$) matters.

3. Whereas the FAQL/BQL model is independent of the discount factor in single-state environments, actual independent Q-learning dynamics are clearly influenced by changes in $\gamma$ and exhibit fundamentally different behaviour.

## 3 A CHOICE-PROBABILITY-AWARE MODEL OF INDEPENDENT Q-LEARNING

The discrepancies between the FAQL/BQL models and independent Q-learning arise from the implicit assumption that *all* Q-values are updated at each step. Recently Hu et al. (2022) proposed an adjusted "continuity equation model" of independent Q-learning in large-scale multi-agent systems modelled as population games. However, their model is limited to the case $\gamma = 0$. Thus, we cannot apply it to explain all of the stylised discrepancies from above.

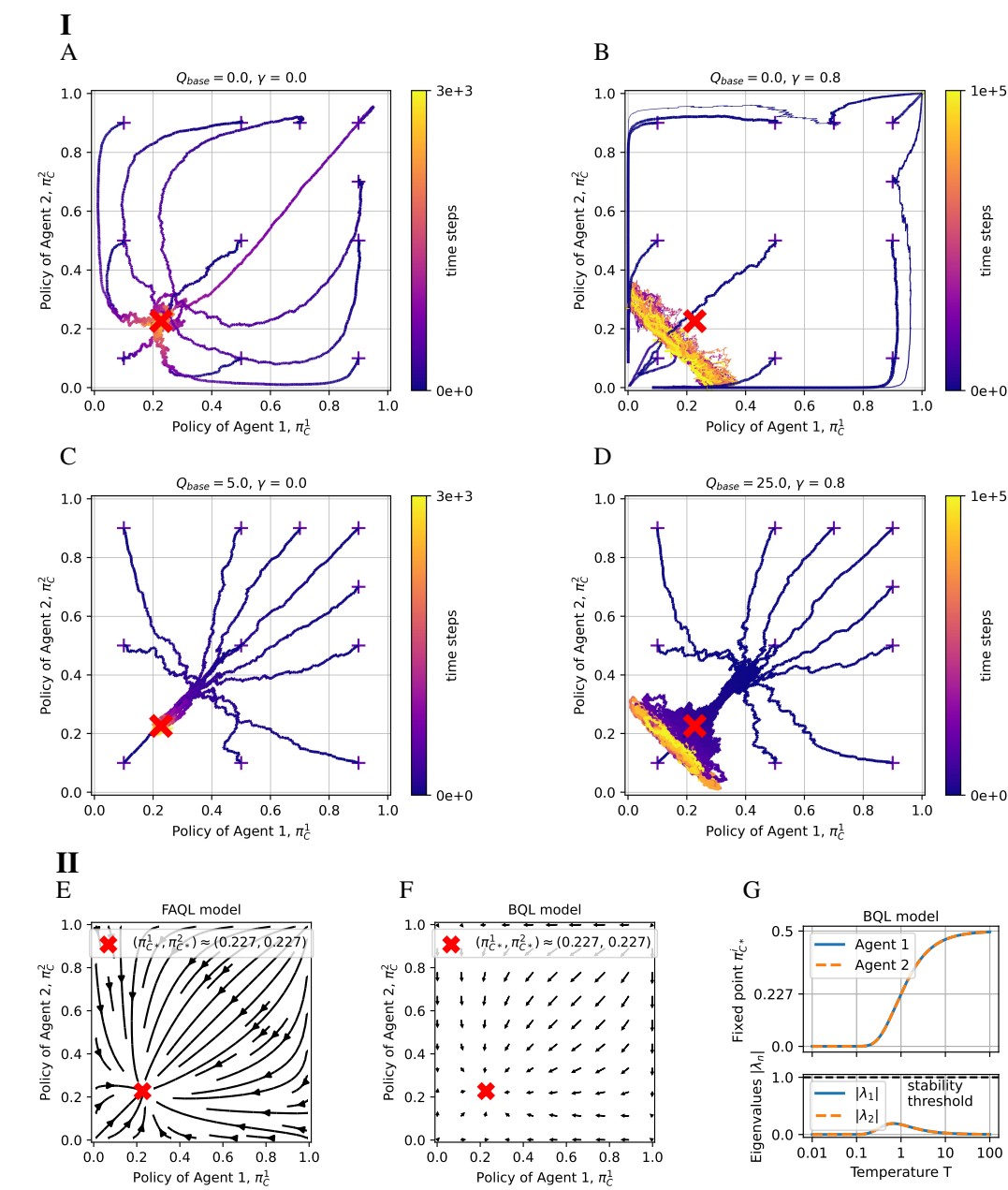

Figure 2: Comparison between averaged policy trajectories of independent Q-learning on the Prisoner's Dilemma (**I**) and previous deterministic models (**II**) for $T = 1$ and $\alpha = 0.01$. **I**: Top panels (A, B): $Q_{base} = \min(\mathbf{R})/(1-\gamma)$. Bottom panels (C, D): $Q_{base} = \max(\mathbf{R})/(1-\gamma)$. Left panels (A, C): $\gamma = 0$. Right panels (B, D): $\gamma = 0.8$. For each initialisation, five runs are executed. Trajectories from the same initialisation are grouped based on their final location in policy space (below or above the diagonal from (0,1) to (1,0)), and the mean of each group is plotted. Line thickness indicates the proportion of runs in each group. The colour gradient (purple to yellow) indicates time evolution. The red cross marks the fixed point of the FAQL/BQL model. Note that in panel B, some trajectories initialised in the top right *appear* to converge to the metastable phase of mutual cooperation in the depicted time span of $1 \times 10^5$ steps. **II**: Vector fields of previous models. E: FAQL model in continuous time, defined by equation 4. F: BQL model in discrete time, defined by equation 11. G: Stability analysis of the BQL model (see appendix C). It has a unique symmetric fixed point $\boldsymbol{\pi}_* > 0$, depending on the temperature $T > 0$. All absolute eigenvalues of the Jacobian at $\pi_{C*}^i$ are below 1, indicating a stable node.

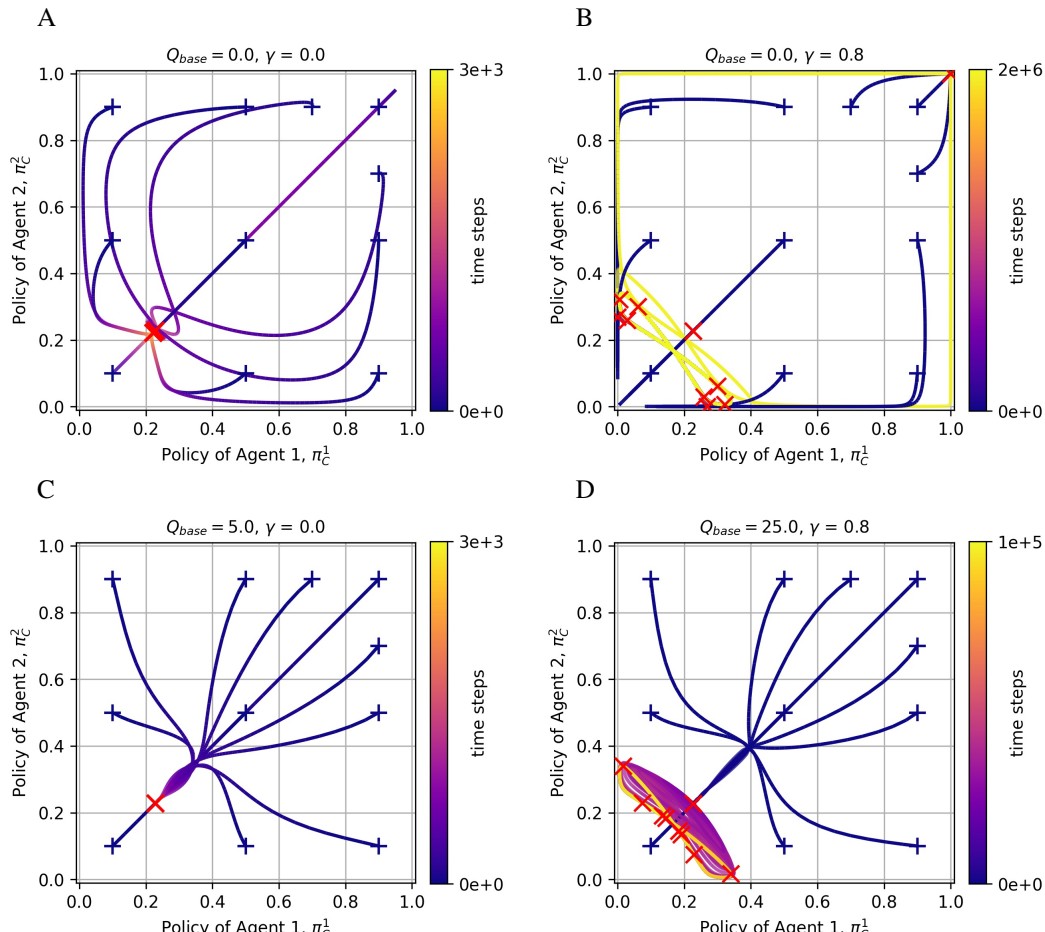

Figure 3: Projection of the 4D deterministic dynamics, defined by equation 6, into 2D policy space for the Prisoner's Dilemma with $T = 1$, $\alpha = 0.01$, and different values of $\gamma$ and $Q_{base}$. The colour gradient (purple to yellow) represents time evolution. The end point of each trajectory is indicated by a red cross. Top panels (A, B): $Q_{base} = \min(\mathbf{R})/(1 - \gamma)$. Bottom panels (C, D): $Q_{base} = \max(\mathbf{R})/(1 - \gamma)$. Left panels (A, C): $\gamma = 0$. Right panels (B, D): $\gamma = 0.8$. Note that in panel B, the trajectory initialised at $\pi_C^i(0) = 0.9$ eventually converges to the fixed point $\pi_{C*}^i \approx 0.227$, but only after $4 \times 10^7$ steps, far beyond the depicted $2 \times 10^6$ steps.

Here, we propose a more accurate approximation model for independent Q-learning in a single-state, repeated environment, with discounting but no memory, that correctly makes update frequencies proportional to current policies and can explain all of the dynamics observed above.

Our model retains the discrete time steps of actual Q-learning and represents the expected change in Q values from step to step:

$$
\mathbb{E}_{\mathbf{A}(t)}[Q_{a^i}^i(t+1) \mid Q^i(t)] = Q_{a^i}^i(t) + \alpha \pi_{a^i}^i(t) \left( \mathbb{E}_{A^{-i}(t)} R_{a^i A^{-i}(t)}^i + \gamma \max_{b^i \in \mathcal{A}^i} Q_{b^i}^i(t) - Q_{a^i}^i(t) \right).
$$
(6)

It is *not possible* to reduce these dynamics to policy space since $\mathbb{E}\Delta Q^i(t + 1)$ depends on absolute Q values rather than only on Q differences.

However a *projection* of the dynamics onto policy space can still reveal interesting dynamics (Figure 3). For reasonably small learning rates ($\alpha = 0.01$), a comparison with the averaged trajectories of actual independent Q-learning (figure 2) demonstrates that our model captures the observed complexities. For $\gamma = 0$, all trajectories converge to the fixed point of the FAQL/BQL model,

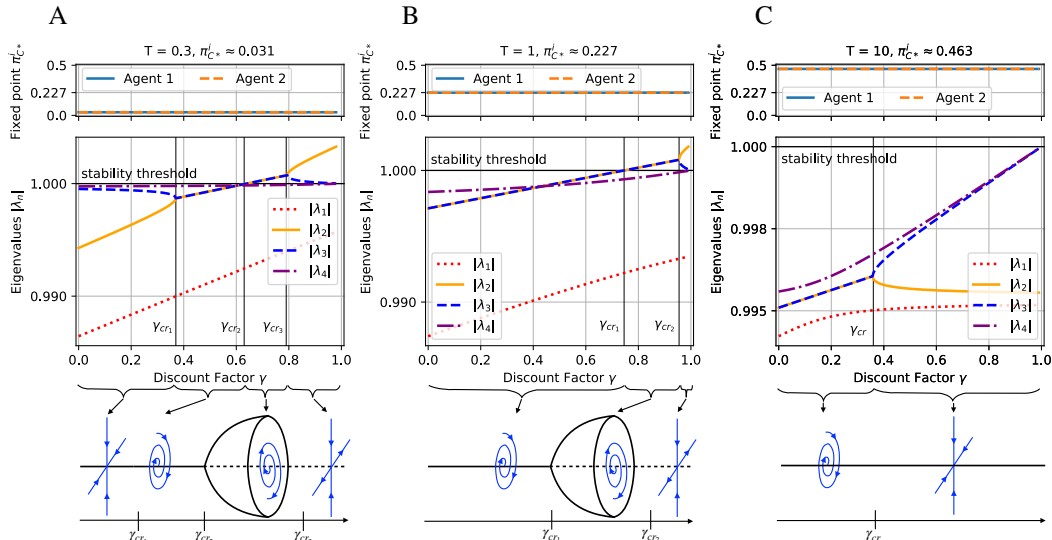

Figure 4: Stability analysis of equation 6 for the Prisoner's Dilemma, with $\alpha = 0.01$ and three different temperature values: $T = 0.3$ (A), $T = 1$ (B), and $T = 10$ (C). The first row shows a projection of the 4D fixed point $\mathbf{Q}_*$, defined by equation 20, in 2D policy space, illustrating how the equilibrium policy is not affected by the discount factor. The second row shows the absolute eigenvalues of the Jacobian at $\mathbf{Q}_*$ as a function of $\gamma$, with the discrete-time stability threshold ($|\lambda| = 1$) highlighted. It demonstrates that although the position of the fixed point in policy space remains unaffected by $\gamma$, its stability properties in Q space change. The third row provides schematic representations in a lower-dim. representation of the corresponding dynamical regimes for different ranges of $\gamma$, illustrating transitions between stability, oscillatory dynamics, and divergence.

$\pi^i_{C*} \approx 0.227$. In contrast, for $\gamma = 0.8$, the behaviour depends on the initial policies: symmetric initial policies converge to $\pi^i_{C*}$ while asymmetric initial policies lead to oscillatory dynamics. Note that for $Q_{base} = 0, \gamma = 0.8$, the trajectory starting at the symmetric initial condition $\pi^i_C = 0.9$ remains in the cooperation state for up to two million steps, seemingly contradicting the statement just made. However, after an astonishing four *billion* steps, it finally converges to $\pi^i_{C*}$.

These phenomena are readily explained and proven through a stability analysis (Appendix D), offering an efficient approach while avoiding ambiguities of interpreting individual trajectories or specific parameter cases. Note that a policy space projection $\boldsymbol{\pi}_* = \boldsymbol{\pi}(\mathbf{Q}_*)$ of a fixed point $\mathbf{Q}_*$ of equation 6 is also an equilibrium solution of the FAQL/BQL model, *but their stability differs*. While $\boldsymbol{\pi}_*$ is a stable node for all values of $T$ and $\gamma$ in the FAQL/BQL model, the stability of $\mathbf{Q}_*$ is more nuanced.

For $T = 1$, the 4D fixed point $\mathbf{Q}_*$ is a stable focus for $\gamma \lesssim 0.75$ (figure 4). At $\gamma_{cr_1} \approx 0.75$, the system undergoes a supercritical Neimark–Sacker bifurcation, turning the stable focus into an unstable one, around which a stable limit cycle emerges. All trajectories with asymmetric initial conditions, even with minimal deviation, converge to this limit cycle, leading to the oscillations observed for $\gamma = 0.8$. For $\gamma \gtrsim 0.95$, the unstable focus turns into a saddle node. Figure 5 illustrates these different dynamical regimes. Note that the trajectory initialised at $\pi^i_C(0) = 0.9$ for $\gamma = 0.97$ remains at mutual cooperation ($\pi^i_C \approx 1$) within any finite number of steps feasible for computational simulation. However, the equations show that this is *not* a stable fixed point.

Using the fixed point equation 20 (Appendix D), which also defines an agent's target values given its opponent's policy, we can now explain the observed phenomena step by step. As a representative case, we analyse the trajectory shown in Fig. 1C.

**Metastable Phases**   Starting at zero, all $Q$-values grow at first. Since defection yields higher rewards, $Q^i_D$ grows faster than $Q^i_C$ so that $\Delta Q^i$ increases and both cooperation probabilities decline fast. Assuming w.l.o.g. $\pi^2_C(0) < \pi^1_C(0)$, the cooperation probability of agent 2 approaches zero

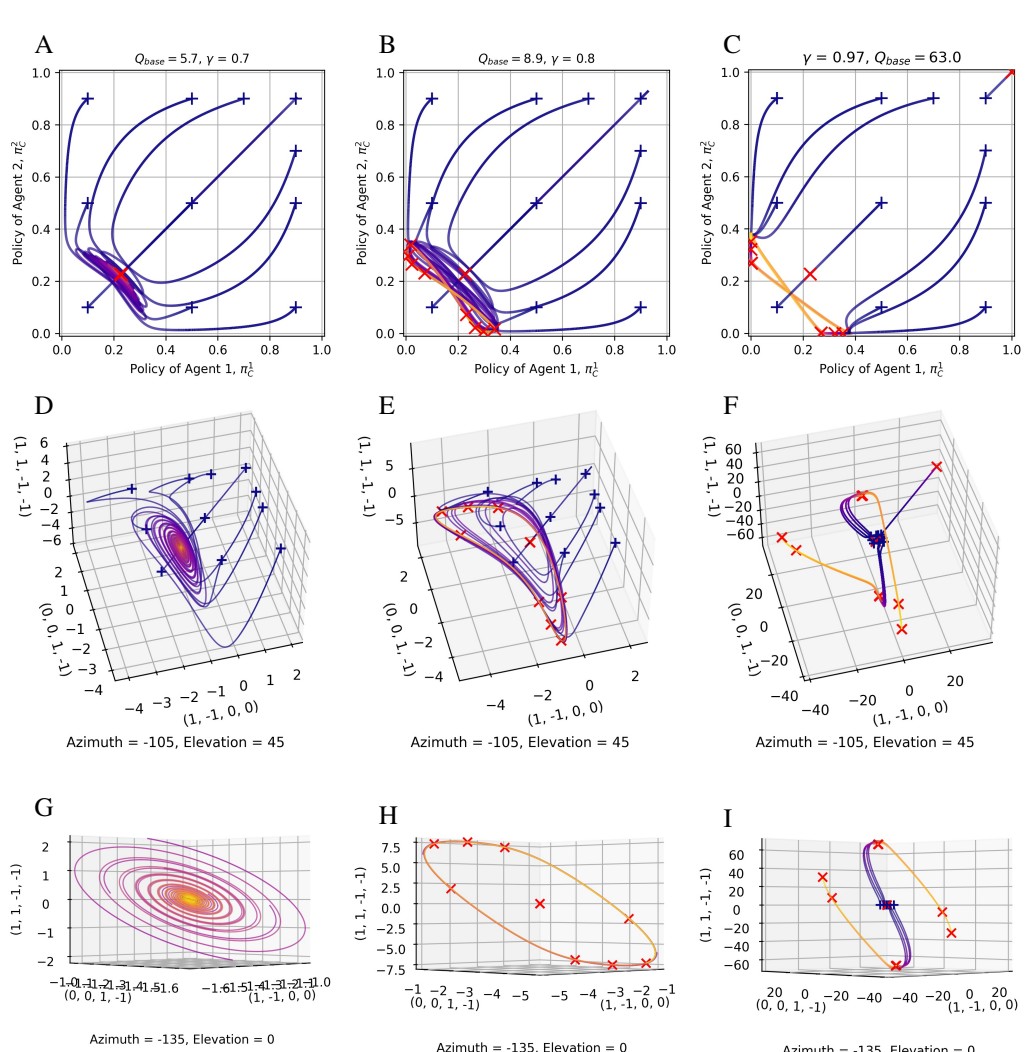

Figure 5: Projection of the 4D deterministic dynamics, defined by equation 6, for the Prisoner's Dilemma with $T = 1$, $\alpha = 0.01$ and different values of $\gamma$. Left panels (A, D, G): $\gamma = 0.7$. Middle panels (B, E, F): $\gamma = 0.8$. Right panels (G, H, I): $\gamma = 0.97$. All trajectories are initialised around the fixed point $Q$-values, defined by equation 20: $Q_{base} = Q_{C*} + \Delta Q_*/2$. The colour gradient (purple to yellow) represents time evolution over $3 \times 10^4$ steps. The end point of each trajectory is indicated by a red cross. Top panels (A, B, C): Projection of 4D dynamics into 2D policy space. Middle panels (D, E, F): Projection into a 3D space defined by the basis vectors $\mathbf{q}_1 = (1, -1, 0, 0)$, $\mathbf{q}_2 = (0, 0, 1, -1)$, and $\mathbf{q}_3 = (1, 1, -1, -1)$. The first two dimensions represent the $\Delta Q^i$-values, while the third dimension captures the difference between agents. Bottom panels (G, H, I): Projection into the same 3D space, viewed from a different angle. For $\gamma = 0.7$ and $\gamma = 0.8$, only the last two-thirds of the time evolution are shown for clarity. For $\gamma = 0.7$, the unique fixed point $\pi_{C*}^i$ is a stable focus. For $\gamma = 0.8$, it is an unstable focus surrounded by a stable limit cycle for all asymmetric joint policies. For $\gamma = 0.97$, it is a saddle point, with stable eigenvectors projected onto the diagonal of the policy space and unstable eigenvectors directed perpendicular to it. The trajectory initialised at $\pi_C^i(0) = 0.9$ remains at mutual cooperation ($\pi_C^i \approx 1$) within any finite number of steps feasible for computational simulation. Note however that the equations show that this is *not* a true fixed point and pure policies are prohibited due to $T > 0$.

first. At this point agent 1's target values given $\pi_C^2 \approx 0$ are according to equation 20

$$Q_{C,target}^1 \approx 0 + \frac{\gamma}{1-\gamma} = 4, \qquad Q_{D,target}^1 \approx 1 + \frac{\gamma}{1-\gamma} = 5,$$

resulting in $\pi_C^1 \approx 0.27$. In return, agent 2's target values are then $Q_{C,target}^2 \approx 9.1$ and $Q_{D,target}^2 \approx 10.4$. Since agent 2 primarily defects, $Q_D^2$ updates frequently and reaches its target quickly, while $Q_C^2$ lags due to very infrequent updates, exacerbated by the Boltzmann policy's exponential amplification of Q differences, thus keeping $\pi_C^2$ near zero. This metastable phase persists until $Q_C^2$ finally received enough updates as well to approach its target. Over time, $Q_C^2$ gradually catches up, closing the gap $\Delta Q^2$, and the assumption $\pi_C^2 \approx 0$ no longer holds.

**Oscillations**  As $\pi_C^2$ grows, the expected rewards and hence also the target values of agent 1 grow. But again, due to the asymmetric update frequency, $Q_D^1$ increases much faster than $Q_C^1$. The policy $\pi_C^1$ plummets close to zero. This has the effect that the target values of agent 2 decrease drastically as well, closing the $\Delta Q^2$ gap even further. As a result, $\pi_C^2$ grows rapidly. Now, the roles of agent 1 and 2 are swapped and the process begins all over again, albeit with a shorter period. An oscillating pattern emerges. The oscillations can be understood as a feedback loop in which the agents' adaptations consistently lag behind the changes of their effective environment. This phenomenon, known as the 'moving target problem' (Sutton & Barto, 2018), poses a significant challenge in MARL (Albrecht et al., 2024; Hernandez-Leal et al., 2017).

## 4 DISCUSSION AND CONCLUSION

Our analysis underscores the importance of accounting for $Q$-value update frequencies to understand independent Q-learning dynamics. Our deterministic approximation captures behaviours that simpler policy-space approximations cannot describe. This is evident in the Prisoner's Dilemma, where dynamics may oscillate after long metastable transients, while the FAQL/BQL models predict global convergence. While our focus here was on single-state environments, our insights are also relevant in cyclic multi-state environments. Preliminary results show that in environments with two states and low transition probabilities, similar behaviour can emerge.

We illustrated how a Boltzmann policy can cause update frequencies to approach zero, inducing metastable phases. Their length can far exceed any realistic number of learning steps, making them easy to mistake for equilibria. This highlights the importance of examining all dynamic variables (all $Q$-values) rather than focusing solely on the target variables of interest (the policy), as only a few of these might display perceptible drift during a metastable phase indicating instability (Kittel et al., 2017). Apparent stable cooperation in the Prisoner's Dilemma is only a prolonged transient of the Q-learning process that eventually converges to mutual defection. While such misinterpretations are relatively easy to avoid in simple environments like the Prisoner's Dilemma, they become far more challenging in complex environments with many agents, actions, and multiple Nash equilibria.

Further, we demonstrated how a moving target problem can cause stable oscillations that prevent convergence. Although tweaks like batch learning, frequency-adjusted updates, or alternative exploration policies (epsilon-greedy) can help mitigate these symptoms, the root cause of oscillations lies in the *non-stationarity* of the effective environment for each agent, a fundamental challenge in MARL (Hernandez-Leal et al., 2019).

While the proposed model improves to capture the dynamics of independent Q-learning, it cannot predict exact timing or outcomes of individual runs, which depend on randomness and sensitivity to initial conditions. Incorporating a noise term—turning the ordinary difference equations into stochastic ones—could improve fidelity in this regard.

If the independent Q-learning algorithm with a Boltzmann policy is used as a model of actual learning processes occurring in humans or other organisms, the described complex dynamics should be considered interesting features worthy of further study. Most of the time, however, MARL algorithms are used as a numerical tool for finding certain types of strategic equilibria. For that application, the described dynamics should rather be considered a bug than a feature. In that context, addressing the non-stationarity challenge is crucial for developing scalable MARL algorithms with robust convergence guarantees, which remains an open research problem (Albrecht et al., 2024). As demonstrated, a dynamical systems perspective can be helpful for future work in this regard.

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

---

**Algorithm 1:** Independent Q-learning with Boltzmann policy in a single-state environment

---

**Input:** Action space $\mathcal{A}^i$, learning rate $\alpha^i$, discount factor $\gamma^i$, temperature parameter $T^i$ for each agent $i$, common environment E
**Output:** Learned $Q$-values $Q^i_{a^i}$ for each agent $i$
Initialise $Q^i_{a^i}$ arbitrarily for all $a^i \in \mathcal{A}^i$ for each agent $i$
**while** *not reached terminal time step* **do**
    **for** *each agent $i$* **do**
        Choose action $a^i$ with Boltzmann policy:
$$\pi^i_{a^i} \leftarrow \frac{e^{Q^i_{a^i}/T^i}}{\sum_{b^i} e^{Q^i_{b^i}/T^i}} \text{ for all } a^i \in \mathcal{A}^i$$
$$a^i \sim \pi^i_{a^i}$$
    **end**
    Take joint action $\mathbf{a} = (a^1, a^2, ..., a^n)$ in the environment E
    **for** *each agent $i$* **do**
        Observe own reward $r^i$ in the environment E
        Update $Q$-value of chosen action:
$$Q^i_{a^i} \leftarrow Q^i_{a^i} + \alpha^i \left[ r^i + \gamma^i \max_{b^i} Q^i_{b^i} - Q^i_{a^i} \right]$$
    **end**
**end**

---

## A   Q-LEARNING

The original *single-agent* incremental Q-learning algorithm (Watkins & Dayan, 1992) is defined in the framework of a finite Markov Decision Processes (MDP) (Albrecht et al., 2024), consisting of a finite non-empty set of states $\mathcal{S}$, a subset of terminal states $\mathcal{S}_{terminal} \subset \mathcal{S}$, a finite non-empty set of actions $\mathcal{A}$, a reward function $R : \mathcal{S} \times \mathcal{A} \times \mathcal{S} \to \mathbb{R}$ and a state transition probability function $T : \mathcal{S} \times \mathcal{A} \times \mathcal{S} \to [0, 1]$, such that for all $s \in \mathcal{S}, a \in \mathcal{A} : \sum_{s' \in \mathcal{S}} T(s'|a, s) = 1$.

At each time step $t$, a singular Q-learning agent observes state $S(t) = s$ of the environment, chooses action $A(t) = a$, upon which the environment transitions to state $S(t + 1) = s_{next}$ and the agent receives the reward $R(s, a, s_{next}) = r$. The agent then updates its value estimate of the state-action pair $(s, a)$, called $Q$-value, via the update rule

$$Q_{s,a}(t + 1) = Q_{s,a}(t) + \alpha \left[ r + \gamma \max_{b \in \mathcal{A}} Q_{s_{next},b}(t) - Q_{s,a}(t) \right], \tag{7}$$

$$Q_{s',a'}(t + 1) = Q_{s',a'}(t) \qquad \text{for all } (s', a') \neq (s, a), \tag{8}$$

where $\alpha \in [0, 1)$ is the agent's *learning rate*, and the *discount factor* $\gamma \in [0, 1)$ determines the weight the agent assigns to the current estimate of the optimal value of the next state $s_{next}$. Note that *only* the $Q$-value of the state-action pair actually played at time $t$ gets updated, the remaining $Q$-values retain their current values. Q-learning is guaranteed to converge to optimal state-action values under certain conditions (Watkins & Dayan, 1992), with one key requirement being that the environment remains stationary—a crucial property of an MDP.

### A.1   REMARKS

Formally, the environment analysed in this work consists of a single *non-terminal* state, $s_0$, defined by equation 1. After each non-terminal time step, the environment transitions back to $s_0$. The learning process concludes at a terminal time step, at which point the environment transitions to a *terminal* state, $s_{terminal}$. By definition, no rewards are provided in the terminal state, and agents remain there indefinitely (Albrecht et al., 2024). We note that this setup corresponds to the game-theoretic definition of a finitely repeated normal-form game, where agents do *not* condition their policies on past interactions.

In the original single-agent Q-learning algorithm, the discount factor $\gamma^i$ is a hyperparameter that determines an agent's preference for future state values in *multi-state* environments. The necessity of including a discount factor in a *single-state* environment, as considered here, is therefore debatable. Some studies effectively set $\gamma^i = 0$ by defining the environment to transition into a terminal state after each round (Galstyan, 2013; Kianercy & Galstyan, 2012; Leonardos & Piliouras, 2022; Hu et al., 2022). Others define the environment as static yet repetitive and keep the term involving $\gamma$ (Tuyls et al., 2003; Babes et al., 2009; Wunder et al., 2010; Kaisers & Tuyls, 2010; Zschache, 2018; Mintz & Fu, 2024). To preserve the algorithm's core structure—where the term involving $\gamma^i$ is a defining feature—we consider a repetitive environment and retain the discount factor. Given that the agents lack knowledge of when the game will end, our framework is consistent with the common interpretation of $\gamma^i$ to be the agent's belief about the probability that the game continues in the next time step.

The Boltzmann exploration policy is chosen over common alternatives like epsilon-greedy because it uses a smooth probability distribution based on $Q$-values rather than discrete choices. Some studies suggest this mechanism aligns with human and animal decision-making in competitive and observational learning tasks (Lee et al., 2004; Kim et al., 2009). The temperature parameter $T^i > 0$ regulates the exploration-exploitation trade-off: higher $T^i$ promotes exploration by equalising probabilities, while lower $T^i$ emphasises exploitation of actions with higher $Q$-values. As $T^i \to 0$, the agent converges to a pure policy. We keep the temperature constant throughout the learning process, rather than annealing it (Sandholm & Crites, 1996), to simplify the process and enhance the interpretability of the results.

In QL, the outcome of the learning process is typically interpreted as a pure policy: the action with the maximum Q-value in a given state is regarded as the "learned" action. However, in this work, we focus on the dynamics of the learning process itself, interpreting the Boltzmann distribution as the "learned" policy at any time $t$, as it reflects the agent's probabilistic decision-making process. Our primary interest lies in understanding the long-term behaviour of the learning process as a function of parameters and initial conditions.

## B  THE BQL MODEL

In the BQL model (Barfuss et al., 2019), agents interact $K \in \mathbb{N}_+$ times under a *constant* joint policy $\boldsymbol{\pi}(t)$. Information from these interactions are stored inside a batch of size $K$. At the update step $(t + K)$, agents then use the sample average of the gathered experience to update their $Q$-values and subsequently the joint policy $\boldsymbol{\pi}(t + K)$. With a minor abuse of notation to improve readability, equation 2 is modified to

$$Q_{a^i}^i(t + K) = Q_{a^i}^i(t) + \alpha^i D_{a^i, \mathbf{A}(t), ..., \mathbf{A}(t+K), Q^i(t)}^i, \tag{9}$$

$$D_{a^i, \mathbf{A}(t), ..., \mathbf{A}(t+K), Q^i(t)}^i := \frac{1}{K_{a^i}} \sum_{k=0}^{K-1} \delta_{A^i(t+k)a^i} \left[ R_{\mathbf{A}(t+k)}^i + \gamma^i \max_{b^i \in \mathcal{A}^i} Q_{b^i}^i(t) - Q_{a^i}^i(t) \right], \tag{10}$$

where $K_{a^i} := \max\left(1, \sum_{k=0}^{K-1} \delta_{A^i(t+k)a^i}\right)$ denotes the number of times agent $i$ played action $a^i$. To avoid division of zero, $K_{a^i} := 1$ if the action $a^i$ was never played. For a batch size of $K = 1$, batch Q-learning is equal to regular Q-learning. Note however that for $K > 1$, batch learning allows to update *multiple* Q-values per agent per update step—all Q-values whose actions were played in the batch.

In the infinite batch limit $K \to \infty$ (and subsequently $K_{a^i} \to \infty$), the stochastic batch temporal difference error equation 10 becomes almost surely (a.s.) deterministic due to the law of large numbers. The limit implies that, with probability one, *all* Q-values are updated simultaneously at each update step. This enables the derivation of a deterministic update rule in a separated *update timescale* $u$ that operates exclusively in the lower-dimensional policy space (see appendix B.1 for a detailed derivation):

$$\pi_{a^i}^i(u + 1) = \frac{\pi_{a^i}^i(u) \exp[\alpha^i D_{a^i \boldsymbol{\pi}(u)}^i / T^i]}{\sum_{b^i \in \mathcal{A}^i} \pi_{b^i}^i(u) \exp[\alpha^i D_{b^i \boldsymbol{\pi}(u)}^i / T^i]}, \tag{11}$$

where

$$D_{a^i, \boldsymbol{\pi}(u)}^i := \mathbb{E}_{A^{-i}(u) \sim \pi^{-i}(u)} R_{a^i A^{-i}(u)}^i - T^i \ln \pi_{a^i}^i(u). \tag{12}$$

Note that for single-state environments, all terms which include the discount factor $\gamma^i$ vanish in the derivation of equation 11. As in the FAQL model, this is again due to the implicit assumption that all $Q$-values get updated simultaneously. Good agreement of equation 11 with actual behaviour for $K \approx 10^3 - 10^4$ was demonstrated in Barfuss (2022), but not for smaller $K$-values. To emphasise its distinction from incremental Q-learning, we will refer to this model throughout this work as the 'Batch Q-Learning' (BQL) model. In single-state environments, the FAQL model corresponds to the continuous-time limit of the BQL model; hence, we also collectively refer to them as the 'FAQL/BQL model'.

A fixed point policy $\pi_*$ of equation 11 can be determined by finding the roots of equation 12 for all $i, a^i$. After normalisation, this results in the two-dimensional system of equations

$$\pi^i_{a^i*} = \frac{\exp[\mathbb{E}_{A^{-i} \sim \pi_*^{-i}} R^i_{a^i A^{-i}}/T]}{\sum_{b^i \in \mathcal{A}^i} \exp[\mathbb{E}_{A^{-i} \sim \pi_*^{-i}} R^i_{b^i A^{-i}}/T]}. \tag{13}$$

This equation can also be interpreted outside the learning context as defining a "soft" version of Nash equilibrium based on a form of bounded rationality rather than full rationality: if the equation is fulfilled, both players do not maximise but "soft maximise" their reward under the correct assumption that the other player does likewise, by playing the corresponding Boltzmann policy. In behavioural game theory, this form of equilibrium is called 'Logit Quantal Response equilibrium' (McKelvey & Palfrey, 1995). As experimental evidence from humans suggest that indeed boundedly rational human decisions sometimes approximate such soft equilibria (Teeselink et al., 2024), the question of whether MARL algorithms converge to such points as well is an important plausibility check.

## B.1 DERIVATION OF THE BQL EQUATIONS

In the limit $K \to \infty$ (and subsequently $K_{a^i} \to \infty$), the stochastic batch temporal difference error equation 10 becomes almost surely (a.s.) deterministic because of the law of large numbers. It can be written in dependence of all $Q$-values at time t as

$$
\begin{aligned}
D^i_{a^i, \mathbf{Q}(t)} &:= \lim_{K \to \infty} D^i_{a^i, \mathbf{A}(t), \ldots, \mathbf{A}(t+K), Q^i(t)} \\
&\overset{\text{a.s.}}{=} \mathbb{E}_{A^i(t)=a^i, A^{-i} \sim \pi^{-i}(t)} \left( \delta_{A^i(t) a^i} \left[ R^i_{\mathbf{A}(t)} + \gamma^i \max_{b^i \in \mathcal{A}^i} Q^i_{b^i}(t) - Q^i_{a^i}(t) \right] \right) \\
&= \mathbb{E}_{A^{-i}(t) \sim \pi^{-i}(t)} R^i_{a^i A^{-i}(t)} + \underbrace{\gamma^i \max_{b^i \in \mathcal{A}^i} Q^i_{b^i}(t)}_{\text{constant in } a^i} \\
&\quad - T^i \ln \pi^i_{a^i}(t) - \underbrace{T^i \ln \sum_{b^i \in \mathcal{A}^i} \exp[Q^i_{b^i}(t)/T^i]}_{\text{constant in } a^i},
\end{aligned}
\tag{14}
$$

where the last two terms are the inverse of equation 3. The deterministic update rule for the $Q$-values in the separated *update* timescale $u$ then reads

$$Q^i_{a^i}(u+1) = Q^i_{a^i}(u) + \alpha^i D^i_{a^i, \mathbf{Q}(u)}. \tag{15}$$

Inserting equation 15 into equation 3 returns a deterministic update rule for the policy,

$$
\begin{aligned}
\pi^i_{a^i}(u+1) &= \frac{\exp[Q^i_{a^i}(u+1)/T^i]}{\sum_{b^i \in \mathcal{A}^i} \exp[Q^i_{b^i}(u+1)/T^i]} \\
&= \frac{\exp[Q^i_{a^i}(u)/T^i] \exp[\alpha^i D^i_{a^i, \mathbf{Q}(u)}/T^i]}{\sum_{b^i \in \mathcal{A}^i} \exp[Q^i_{b^i}(u)/T^i] \exp[\alpha^i D^i_{b^i, \mathbf{Q}(u)}/T^i]} \\
&= \frac{\pi^i_{a^i}(u) \exp[\alpha^i D^i_{a^i, \mathbf{Q}(u)}/T^i]}{\sum_{b^i \in \mathcal{A}^i} \pi^i_{b^i}(u) \exp[\alpha^i D^i_{b^i, \mathbf{Q}(u)}/T^i]}.
\end{aligned}
\tag{16}
$$

As it is, equation 16 depends on the four-dimensional vector $\mathbf{Q}(u)$. To have an approximation that conveniently reduces the learning dynamics to the two-dimensional policy space, equation 16 needs

to be expressed purely in terms of $\boldsymbol{\pi}(u)$. Luckily, one can make use of the fact that equation 16 is invariant under adding terms to $D^i_{a^i,\mathbf{Q}(u)}$ that are constant in $a^i$, such as the last term of equation 14. Note that in single-state environments, also the second term including the discount factor is constant in action—no matter which actions the agents choose, the environment transitions back to the same unique non-terminal state—and can thus be excluded. This means that the dynamics are *independent* of the discount factor. Equation 16 simplifies to

$$\pi^i_{a^i}(u+1) = \frac{\pi^i_{a^i}(u)\exp[\alpha^i D^i_{a^i,\boldsymbol{\pi}(u)}/T^i]}{\sum_{b^i\in\mathcal{A}^i}\pi^i_{b^i}(u)\exp[\alpha^i D^i_{b^i,\boldsymbol{\pi}(u)}/T^i]}, \tag{17}$$

where

$$D^i_{a^i,\boldsymbol{\pi}(u)} := \mathbb{E}_{A^{-i}(u)\sim\pi^{-i}(u)}R^i_{a^i A^{-i}(u)} - T^i\ln\pi^i_{a^i}(u). \tag{18}$$

## C  STABILITY ANALYSIS OF THE BQL MODEL

We solve the two-dimensional system of equations 13 numerically using the `fsolve` function from Python's SciPy library. For the Prisoner's Dilemma, there exists a unique symmetric fixed point (see figure 2). To determine its stability, we conduct a linear stability analysis at the fixed point. To this end, we calculate the Jacobian

$$J = \begin{pmatrix} \partial_{\pi^1_C}\pi^1_C & \partial_{\pi^2_C}\pi^1_C \\ \partial_{\pi^1_C}\pi^2_C & \partial_{\pi^2_C}\pi^2_C \end{pmatrix} = \begin{pmatrix} 0 & -\frac{p^1_{\pi^2}q^1_{\pi^2}}{T[p^1_{\pi^2}+q^1_{\pi^2}]^2} \\ -\frac{p^2_{\pi^1}q^2_{\pi^1}}{T[p^2_{\pi^1}+q^2_{\pi^1}]^2} & 0 \end{pmatrix}, \tag{19}$$

where

$$p^i_{\pi^{-i}} := \exp[\mathbb{E}_{A^{-i}\sim\pi^{-i}}R^i_{a^i=C,A^{-i}}/T],$$
$$q^i_{\pi^{-i}} := \exp[\mathbb{E}_{A^{-i}\sim\pi^{-i}}R^i_{a^i=D,A^{-i}}/T].$$

Note that the prefactor $-1$ in equation 19 comes from the reward structure in equation 1

$$R^i_{a^i=C,a^{-i}=C} - R^i_{a^i=C,a^{-i}=D} - R^i_{a^i=D,a^{-i}=C} + R^i_{a^i=D,a^{-i}=D} = 3 - 0 - 5 + 1 = -1$$

We calculate the Eigenvalues $\lambda_n$ of the Jacobi matrix numerically with the function `numpy.linalg.eig` from Python's NumPy library. Since all eigenvalues are $|\lambda_n| < 1$, we deduce the discrete-time fixed point to be a stable node.

## D  STABILITY ANALYSIS OF OUR MODEL

The four-dimensional fixed point $\mathbf{Q}_*$ of equation 6 is obtained by finding the roots of the second term for all $i, a^i$. The coupled equations read

$$Q^i_{a^i*} := \mathbb{E}_{A^{-i}\sim\pi^{-i}_*}R^i_{a^i A^{-i}} + \gamma\max_{b^i\in\mathcal{A}^i}Q^i_{b^i*}$$

$$= \mathbb{E}_{A^{-i}\sim\pi^{-i}_*}R^i_{a^i A^{-i}} + \gamma\max_{b^i\in\mathcal{A}^i}\sum_{k=0}^{\infty}\gamma^k\mathbb{E}_{A^{-i}\sim\pi^{-i}_*}R^i_{b^i A^{-i}} \tag{20}$$

$$= \mathbb{E}_{A^{-i}\sim\pi^{-i}_*}R^i_{a^i A^{-i}} + \underbrace{\frac{\gamma}{1-\gamma}\max_{b^i\in\mathcal{A}^i}\mathbb{E}_{A^{-i}\sim\pi^{-i}_*}R^i_{b^i A^{-i}}}_{\text{constant in }a^i}.$$

Note that in the translation of $\mathbf{Q}_*$ to $\boldsymbol{\pi}_*$ via equation 3, the second term of equation 20 is irrelevant as it is an offset constant in $a^i$ and only the differences of the $Q$-values matter. This means that a fixed point of the dynamics described by equation 17 is also a fixed point of equation 6 in *policy space*, and vice versa. So why does the model described by equation 6 behave so differently compared to the FAQL/BQL model?

The key lies in *stability*. Although both models share the same unique fixed point in policy space, their stability properties differ. While it is a stable node for all values of $T$ and all values of $\gamma$ in the BQL and FAQ model, it is more nuanced in the new model.

If we take into account that for the Prisoner's Dilemma, $Q^i_{C*} < Q^i_{D*}$ holds at the fixed point $\mathbf{Q}_*$, the maximum term of equation 6 reduces to $\max(Q^i_{C*} Q^i_{D*}) = Q^i_{D*}$. We can therefore simplify equation 6 at the fixed point $\mathbf{Q}_*$ to

$$\mathbb{E}_{\mathbf{A}(t) \sim \boldsymbol{\pi}(t)}[Q^i_{a^i}(t+1) \mid Q^i_*(t)] = Q^i_{a^i *}(t)$$
$$+ \alpha \pi^i_{a^i *}(t) \left[ \mathbb{E}_{A^{-i}(t) \sim \pi^{-i}_*(t)} R^i_{a^i A^{-i}(t)} + \gamma Q^i_{D*} - Q^i_{a^i *} \right]. \tag{21}$$

To shorten the notation, we omit the dependencies and the fixed point subscript index $*$ in the following, and make use of the relations

$$\partial_{Q^i_C} \pi^i_C = \partial_{Q^i_D} \pi^i_D = \frac{e^{(Q^i_C + Q^i_D)/T}}{T(e^{Q^i_C/T} + e^{Q^i_D/T})^2},$$
$$\partial_{Q^i_D} \pi^i_C = -\partial_{Q^i_C} \pi^i_C = -\partial_{Q^i_D} \pi^i_D = \partial_{Q^i_C} \pi^i_D.$$

To shorten the notation further, we introduce

$$f^i := \alpha \partial_{Q^i_C} \pi^i_C \left[ \pi^{-i}_C R^i_{a^i=C, a^{-i}=C} + (1 - \pi^{-i}_C) R^i_{a^i=C, a^{-i}=D} + \gamma Q^i_D - Q^i_C \right],$$
$$g^i := \alpha \pi^i_C \partial_{Q^{-i}_C} \pi^{-i}_C \left[ R^i_{a^i=C, a^{-i}=C} - R^i_{a^i=C, a^{-i}=D} \right],$$
$$h^i := \alpha \partial_{Q^i_C} \pi^i_C \left[ \pi^{-i}_C R^i_{a^i=D, a^{-i}=C} + (1 - \pi^{-i}_C) R^i_{a^i=D, a^{-i}=D} - (1 - \gamma) Q^i_D \right],$$
$$k^i := \alpha (1 - \pi^i_C) \partial_{Q^{-i}_C} \pi^{-i}_C \left[ R^i_{a^i=D, a^{-i}=C} - R^i_{a^i=D, a^{-i}=D} \right],$$

and

$$v^i := h^i - \alpha(1 - \gamma)(1 - \pi^i_C) + 1$$

which help to write the Jacobi matrix at the fixed point as

$$J = \begin{pmatrix} f^i - \alpha \pi^i_C + 1 & -f^i + \alpha \gamma \pi^i_C & g^i & -g^i \\ -h^i & v^i & k^i & -k^i \\ g^{-i} & -g^{-i} & f^{-i} - \alpha \pi^{-i}_C + 1 & -f^{-i} + \alpha \gamma \pi^{-i}_C \\ k^{-i} & -k^{-i} & -h^{-i} & v^{-i} \end{pmatrix}. \tag{22}$$

We solve the eigenvalues of the Jacobi matrix at the fixed point equation 20 numerically with the function `numpy.linalg.eig` from Python's NumPy library. The absolute eigenvalues are plotted against the discount factor in figure 4 for three different temperature values, revealing that the 4D dynamics may undergo bifurcations upon changes of the discount factor. Figure 4 further depicts that changes of $T$ not only affect the fixed point position but also the effect of the discount factor $\gamma$ on its stability.

