# OpenReview forum: "Explaining Metastable Cooperation in Independent Multi-Agent Boltzmann Q-Learning – A Deterministic Approximation"
_ICLR.cc/2026/Conference — ICLR 2026 Conference Withdrawn Submission_

### Official Review · Reviewer_mYZ6 · 2025-10-28

**Soundness:** 3
**Presentation:** 2
**Contribution:** 2
**Rating:** 4
**Confidence:** 2

**Summary:**

The authors revisit continuous-time approximations of the independent Q-Learning algorithm for multi-agent reinforcement learning. They argue that the well studied Q-Learning dynamics of Tuyls et al. (2003) and Sato and Crutchfield fall short in two main regards:
1. The dynamics evolve in policy space, which forms a subspace of the full Q-space in which the algorithm iterates.
2. The approximations often ignore the effect that the discount factor has on the asymptotic dynamics of the system.

To show the failure modes of previous models, the authors consider the prisoner's dilemma game and show that the algorithm may exhibit complex dynamics, that the dynamics fail to capture. The authors then introduce a discrete time approximation of Q-Learning which makes clear the dependency on the initial Q-values and the discount factor \gamma.

**Strengths:**

The authors make a clear contribution towards the analysis of Independent Q-Learning in multi-agent systems by showing that the Q-initialisation, which most works ignore, has an impact on the long-term behaviour of the algorithm, and may determine whether the algorithm converges or not. Similarly, the authors show the dependency of the dynamic on the discount factor \gamma, which is not present in the FAQL model.

In my view, the biggest strength of the paper is the counter-example to convergence found in the prisoner's dilemma--a relatively simple game when compared to the likes of Diplomacy or poker. The numerical results presented in the paper are fairly convincing, although it would have been nice to have code attached with the paper to be able to reproduce the results.

**Weaknesses:**

In general, I believe the contributions are strong, and are worth sharing with the community, but I think some work is required to improve the presentation of the results. In particular, while the counter-example (prisoner's dilemma) is an important contribution of the paper, the experiments are rather limited. In particular, it would have been useful to see sweeps over Q_base to understand how often the dynamics reach metastable states. Furthermore, the authors should have presented experiments outside of the prisoner's dilemma. The authors could take inspiration from e.g. Hussain et al. 2023 Stability of Multi-Agent Learning in Competitive Networks: Delaying the Onset of Chaos, in which the game and exploration rates are varied and an empirical boundary between convergent and non-convergent behaviours is obtained. Another approach is in Leonardos and Piliouras (2020) Exploration-Exploitation in Multi-Agent Learning: Catastrophe Theory Meets Game Theory, in which the authors restrict to potential games and show that bifurcations occur as the learning rate is varied.

The authors also do not discuss the relative benefits and drawbacks of the discrete time approach to continuous-time models. In particular, while the former often provides a closer approximation to the original algorithm (as we do not need to take \dt -> 0), the latter is more amenable to analysis using the tools of differential dynamical systems theory. This allows, for example,. for a lyapunov function to be easily determined to show that the system is \emph{globally} stable under the dynamics. Instead the authors opt for a linear stability analysis which, while sound, only tells of convergence in a neighbourhood around the equilibirium. Perhaps a way of motivating their own dynamic would be to select a class of games (e.g. potential) and analyse the predictions of their dynamic in this setting.

Finally, the authors do not consider the impact of the exploration rate T (which is set to 1). It is known from previous works (c.f. Galla and Sanders (2011) Complex dynamics in learning complicated games
 and Hussain et al (2023) On the Stability of Learning in Network Games with Many Players) that the exploration rate plays a critical role in the convergence of Q-Learning. This factor should be explored in this setting also.

**Questions:**

In line 357 you refer to empirical convergence after four billion steps. What is the measure of convergence here? Distance to a known NE/QRE under a given threshold?

---

### Official Review · Reviewer_z3yB · 2025-10-31

**Soundness:** 3
**Presentation:** 2
**Contribution:** 1
**Rating:** 2
**Confidence:** 4

**Summary:**

Summary: The paper studies Q-learning in Prisoner's Dilemma (PD). In particular, the paper considers the continuous time approximation of Q-learning dynamics that is generally considered in the literature and discusses a well-known approximation error occuring between this approximation and the actual, discrete-time Q-learning problem. The discrepancy occurs because the discrete-time model had bandit updates (only the chosen action is updated) whereas the continuous-time approximation updates all Q-values. After the paper illustrates these differences, it goes ahead to propose a better approximation that tracks the updates in the Q-values and shows (through simulations in PD and mathematical analysis in the Appendix) that this model better approximates reality leading to convergence, i.e., more stable outcomes, rather than transient behaviour and oscillations as the "standard" approximation originally proposed and adopted in the literature.

**Strengths:**

The paper presents careful visualisations explaining the problem at hand and proposes an update rule that shows more stable performance. Presentation is generally good.

**Weaknesses:**

The analysis is confined to a 2-player Prisoner's Dilemma game and accordingly, the findings are potentially hard to generalise. Many insights presented in the paper are already known, e.g., this approximation error and how this can be fixed, and although the paper correctly cites these resources, e.g., Kaiser and Tuyls 2010, its additional contribution seems to be only marginal (or at least not clear if more than that). The two/three timescales, these of (individual) adaptation and environmental change, are described in the Sato and Crutchfield 2004/2005 paper (also accurately cited by the authors) giving a justification to the non-bandit update model (many Q-value updates for each policy update - and many policy updates for each environmental update). Regarding presentation, I found the long and descriptive captions of Figures generally helpful, but would have appreciated also more crisp descriptions of the main takeaways to make it easier to parse (although this may be subjective). Also, there is some more reliance to the appendix, e.g., equation (20) for stability analysis, that somewhat compromises the reading experience. Finally, I also missed the definition of a "projection" which is used quite a lot in the paper, but, unless I missed, never explicitly defined. The same holds for other technical jargon, e.g., the Neimark-Sacker bifurcation which is mentioned but never defined nor explained. Nevertheless, I think that these presentation issues are generally fixable. What seems to pose more fundamental concerns and limit the current contribution are the problems mentioned above, i.e., that several insights are already known and that the proposed fix (equation 6) is only tested in PD without formal mathematical results.

Generally, I think that the paper reads more as a note or case-study in a limited setting and would be more appropriate for a venue with more targeted audience.

**Questions:**

- What are the differences of the currently proposed method compared to Kaiser and Tuyls (2010) who also propose a fix to this problem? So, what is the novelty of the current paper?
- Is there a formal mathematical proof that the proposed approximation is more accurate or the evidence comes from the simulations? Is that the stability analysis in Appendix D or is this restricted to the specific PD setting?
- Does the proposed fix generalise to other settings/social dilemmas or more complex (environments with state) environments?
- What is a supercritical Neimark–Sacker bifurcation and why is this the type of bifurcation that the system undergoes? This is mentioned in two places in the paper (abstract and page 7) but no definition/explanation is provided.

---

### Official Review · Reviewer_CyUL · 2025-11-01

**Soundness:** 3
**Presentation:** 3
**Contribution:** 3
**Rating:** 8
**Confidence:** 3

**Summary:**

This paper focuses on the learning dynamics of independent Boltzmann Q-learning in the repeated Prisoner’s Dilemma. It reviews previous deterministic approximation models, such as FAQL and BQL, and points out that they fail to capture the true stochastic learning dynamics. To address this, the authors propose a new deterministic model that tracks the evolution in the Q-value space rather than the policy space. This model successfully reproduces key empirical phenomena observed in simulations, including metastable cooperation due to very low update frequencies of Q-values under the Boltzmann policy, and stable oscillations caused by the moving-target problem.

**Strengths:**

1. The paper proposes a choice-probability-aware discrete-time deterministic model, which explicitly accounts for the coupling between the update frequency of Q-values and the probability of policy selection. This model provides a mathematically closer approximation to actual algorithms, capturing phenomena such as metastable cooperation and oscillation that previous models struggled to describe.

2. The paper provides a comprehensive summary of the characteristics of previous Q-learning dynamic models, offering interesting insight.

3. The paper provides an in-depth analysis of the profound impact that learning parameters have on the Q-learning process. Through comprehensive visualizations and detailed explanations, it elucidates the underlying causes of metastable cooperation and oscillation. These two phenomena are typical manifestations of non-stationarity in MARL, reminding MARL researchers to carefully tune parameters and to be cautious in trusting the results produced by the algorithms.

**Weaknesses:**

1. The dynamic model remains nearly as complex as the original algorithm though theoretically rigorous.

2. The experiments in the paper are conducted only on the two-player, single-state Prisoner’s Dilemma. Although the authors mention having preliminary results in environments with two states and low transition probabilities, they do not present these results in the paper.

3. Missing discussion on some closely related studies:

[1] Collective cooperative intelligence, PNAS 2025.
[2] The best of both worlds in network population games: Reaching consensus and convergence to equilibrium, NeurIPS 2023.
[3] Modelling the Dynamics of Multi-Agent Q-learning: The Stochastic Effects of Local Interaction and Incomplete Information, IJCAI 2022.

**Questions:**

1. Is it necessarily the case that a dynamics model defined on the policy space cannot capture the stochastic nature of the actual Q-learning learning process, and therefore fails to reproduce these evolutionary phenomena?

2. Do these phenomena also emerge in large-scale multi-agent systems employing Boltzmann Q-learning? To what extent does an increase in the number of interactions help reduce stochasticity?

---

### Official Review · Reviewer_6G7G · 2025-11-05

**Soundness:** 3
**Presentation:** 4
**Contribution:** 2
**Rating:** 6
**Confidence:** 3

**Summary:**

This paper investigates why independent multi-agent Q-learning with Boltzmann exploration can exhibit complex behaviors (like persistent oscillations and long-lived cooperation) instead of converging. The authors use a new deterministic discrete-time model of independent Boltzmann Q-learning that explicitly accounts for each action’s update frequency. This model takes into consideration of Q-value updates in expectation, weighted by the current policy probabilities, rather than assuming all Q-values update every step as prior models did. Using the single-state Prisoner’s Dilemma as a case study, they demonstrate that their model reproduces several complex emergent dynamics observed in actual Q-learning (e.g. transient mutual cooperation and metastable oscillatory cycles) which earlier deterministic approximations failed to capture.

**Strengths:**

The contribution is quite refreshing. The paper makes a clear advance by identifying why previous deterministic models (like the frequency-adjusted Q-learning ODE of Tuyls et al. and the Batch Q-learning model of Barfuss et al.) deviate from actual independent Q-learning. This is a rather principled and novel correction – by being aware of choice probabilities, the model preserves the full 4-dimensional Q-value space for two players (instead of collapsing to 2-D policy space as earlier models did) so to capture things like oscillations and metastable phases that simpler policy-space models could not. The authors provide a thorough stability and bifurcation analysis, which is very much appreciated. Overall, I find it a pleasant read.

**Weaknesses:**

A concern is that the study centers almost exclusively on a single-state game (Prisoner’s Dilemma). While this example is well-justified and reveals the phenomena, it remains to be shown how widely the insights apply to more complex or high-dimensional environments. The authors do claim that their insights extend to multi-state settings and mention preliminary results for a two-state environment with rare transitions, but these results are not fleshed out in the paper.

I worry that without at least one additional environment or scenario, the reader must take on faith that similar metastable oscillations and bifurcations occur in larger games. Clarifying the generality of the proposed model is important – for instance, would a three-action per agent game or a stochastic game also exhibit a Neimark–Sacker bifurcation and metastable phases under Boltzmann Q-learning?

The analysis finds that in the single-state PD, the position of the fixed point in policy space (the equilibrium cooperation probability) does not depend on gamma, which is interesting. However, the paper provides little intuition for why the equilibrium policy is gamma-invariant in this setting. Is it due to the specific structure of the payoffs (e.g. the difference in immediate rewards fully determining the equilibrium)? Or does it hold more generally for any single-state game with Boltzmann exploration? The authors may want to clarify this point, and how about multi-state?

**Questions:**

How scalable is the proposed deterministic model and its analysis? The paper hints that similar behavior can occur in a two-state scenario. Would the approach of deriving choice-aware update equations work for general Markov games (with many states), or do state transitions introduce fundamentally new challenges (e.g. a much higher-dimensional system, possibly with multiple coupled limit cycles)? How about more agents?

---

### Note · Authors · 2025-12-03

**Comment:**

We are writing to withdraw our submission.
After reviewing the feedback from our reviewers, we have decided that the paper would benefit from revisions to address the concerns raised. Given the limited time of the revision deadline, we believe it is in the best interest of the quality of our work to withdraw the submission and resubmit an improved version to a future venue.
We sincerely thank the reviewers for their constructive feedback.

**Withdrawal Confirmation:**

I have read and agree with the venue's withdrawal policy on behalf of myself and my co-authors.